

# Implicit motor imagery performance is impaired in people with chronic, but not acute, neck pain

Sarah B. Wallwork[1], Hayley B. Leake[2], Aimie L. Peek[3], G. Lorimer Moseley[2,4] and Tasha R. Stanton[2,4]

[1] University of Canberra Research Institute for Sport and Exercise, University of Canberra, Canberra, Australian Capital Territory, Australia
[2] IIMPACT in Health, University of South Australia, Adelaide, SA, Australia
[3] Faculty of Health Sciences, University of Sydney, Sydney, NSW, Australia
[4] Neuroscience Research Australia, Randwick, NSW, Australia

Corresponding author
Tasha R. Stanton,
tasha.stanton@unisa.edu.au

## ABSTRACT

**Background:** People with chronic neck pain have impaired proprioception (i.e., sense of neck position). It is unclear whether this impairment involves disruptions to the proprioceptive representation in the brain, peripheral factors, or both. Implicit motor imagery tasks, namely left/right judgements of body parts, assess the integrity of the proprioceptive representation. Previous studies evaluating left/right neck judgements in people with neck pain are conflicting. We conducted a large online study to comprehensively address whether people with neck pain have altered implicit motor imagery performance.

**Methods:** People with and without neck pain completed online left/right *neck* judgement tasks followed by a left/right *hand* judgement task (control). Participants judged whether the person in the image had their head rotated to their left or right side (neck task) or whether the image was of a left hand or a right hand (hand task). Participants were grouped on neck pain status (no pain; <3 months—acute; ≥3 months—chronic) and pain location (none, left-sided, right-sided, bilateral). Outcomes included accuracy (primary) and response time (RT; secondary). Our hypotheses—that (i) chronic neck pain is associated with disrupted performance for neck images and (ii) the disruption is dependent on the side of usual pain, were tested with separate ANOVAs.

**Results:** A total of 1,404 participants were recruited: 105 reported acute neck pain and 161 reported chronic neck pain. When grouped on neck pain status, people with chronic neck pain were less accurate than people without neck pain ($p = 0.001$) for left/right neck judgements, but those with acute neck pain did not differ from those without neck pain ($p = 0.14$) or with chronic neck pain ($p = 0.28$). Accuracy of left/right hand judgements did not differ between groups ($p = 0.58$). RTs did not differ between groups for any comparison. When grouped on neck pain location, people were faster and more accurate at identifying right-turning neck images than left-turning neck images, regardless of history or location of pain ($p < 0.001$ for both); people with no pain were more accurate and faster than people with bilateral neck pain ($p = 0.001$, $p = 0.015$) and were faster than those with left-sided neck pain ($p = 0.021$); people with right-sided neck pain were more accurate than people with bilateral neck pain ($p = 0.018$). Lastly, there was a significant interaction between

neck image and side of neck pain: people with right-sided neck pain were more accurate at identifying right-sided neck turning images than people with left-sided neck pain ($p = 0.008$), but no different for left-sided neck turning images ($p = 0.62$). **Conclusions:** There is evidence of impaired implicit motor imagery performance in people with chronic neck pain, which may suggest disruptions to proprioceptive representation of the neck. These disruptions seem specific to the neck (performance on hand images intact) but non-specific to the exact location of neck pain.

## INTRODUCTION

Neck pain affects 10–20% of the population in any given year (*Hoy et al., 2010*) and results in impaired movement and proprioception (i.e., the sense of where your body is located in space). A recent systematic review found that people with chronic neck pain have lower proprioceptive acuity than people without pain (*Stanton et al., 2016*). What remains unclear, however, is where the dysfunction in proprioception lies. While commonly assumed to be attributable to peripheral dysfunction (i.e., disrupted input from proprioceptors in the muscle, ligament, and skin due to injury or nociception), other possible contributions include problems with spinal processing and/or the encoding of proprioceptive data in the brain. In fact, people with neck pain may have intact detection and transmission of proprioceptive input, but have disruptions to the cortical proprioceptive representations that allow for planning, executing and coordinating movements (also termed the 'working body schema' (*Parsons, 2001*)). Given that different treatments can be used to target different proprioceptive impairments, filling this research gap has both mechanistic and therapeutic importance.

Investigating whether impaired cortical proprioceptive representation underlies proprioceptive dysfunction is challenging. Functional neuroimaging can provide key information about changes in primary motor cortex representation of a movement pattern (*Elgueta-Cancino, Schabrun & Hodges, 2018*), or in the cortical activation patterns (including functional connectivity) during tasks such as imagined movements (*Lotze et al., 1999*). However, to date there is no method to image proprioceptive representations and thus, behavioural tasks are used. Implicit motor imagery is the most established method to interrogate the cortical proprioceptive representation and left/right judgements of pictured body parts is the most studied (*Schwoebel et al., 2002*; *Coslett et al., 2010*; *Dey et al., 2012*; *Wallwork et al., 2015*) but not the only approach, for example see (*Moseley & Brugger, 2009*). In left/right judgements, people identify, as quickly and as accurately as possible, whether images of the target body part (e.g., a hand) belongs to the left-side or the right-side of the body. In completing this task, the person mentally manoeuvres their own hand into the posture seen in the picture; a process supported by neuroimaging evidence of activation of motor related areas during the task (*Parsons et al., 1995*; *Michelon, Vettel & Zacks, 2006*). When the target body part resides in the midline, such as

for the neck and back, the task has been adapted and a judgement is made about whether the person in the image has their head or trunk rotated or laterally flexed to the left-side or right-side. Performance (accuracy) in this task is likely to be dependent on an intact neural proprioceptive representation for the target body part, that is the neural representations that coordinate, plan and execute movement (*Bray & Moseley, 2011*), although recent work has questioned whether this occurs as strongly for body parts that reside in the midline—see *Alazmi et al. (2018)*. Notably, because real movement is not permitted during these tasks, peripheral/spinal contributors to proprioception are limited (although see *Silva et al. (2011)*) for evidence that some peripheral contribution still exists), thus the task is thought to primarily target cortical proprioceptive representations.

Previous work evaluating implicit motor imagery performance in people with neck pain has found conflicting evidence. People with recurrent neck pain ($n = 30$) were significantly less accurate than pain-free controls on a neck left/right judgement task (*Elsig et al., 2014*), but people with chronic whiplash associated disorder (WAD) ($n = 64$) were not different from controls (*Pedler, Motlagh & Sterling, 2013*). Additionally, *Elsig et al. (2014)* evaluated performance only on a neck left/right judgement task, making it unclear whether the impaired performance is specific to the painful body part (discrete dysfunction of neck proprioceptive representation), or merely represents impaired spatial performance (regardless of the body part image used) or indeed impaired central nervous system processing. Given this uncertainty and the relatively modest sample sizes of previous work, it is key to evaluate implicit motor imagery performance in people with neck pain using a larger sample to comprehensively explore the presence and nature of any dysfunction.

Here we describe a large, online study recruiting a representative sample of people with neck pain and healthy pain-free controls who performed left/right judgements of neck rotation and of hands. Firstly, we extended past work by investigating the contribution of pain duration and pain location on task performance, as has been investigated in people with and without back pain (*Bowering et al., 2014*). Secondly, we evaluated whether impairment on the left/right judgement task is location- or movement-specific. In people with pathological arm pain, impaired performance on a hand left/right judgement task is specific to images that correspond to their painful hand (*Moseley, 2004b*). It is unknown if a similar effect would occur in people with neck pain (e.g., left sided-pain, impaired performance only on images of left neck rotation), but if present, it would suggest a more nuanced dysfunction in proprioceptive representation than previously realised.

We had two main aims: to determine whether neck pain and its duration are associated with impaired left/right neck rotation judgement performance (Aim 1), and to determine whether the location/side of neck pain is associated with impaired performance for left-turning and right-turning neck rotation images (Aim 2). Within Aim 1, we also evaluated left/right hand judgements to determine whether any impairment in performance in people with neck pain (compared with healthy controls) was specific to the neck. If performance was also reduced in a hand task, this would imply a more global problem—i.e., that any problems with neck motor imagery are not specific to the neck and may therefore simply reflect forced-choice response time impairment (i.e., generally poor

at the task). For Aim 1, we hypothesised that the presence and duration of neck pain would affect performance in a left/right neck rotation judgement task, but not a left/right hand judgement task. Specifically, we hypothesised that people with chronic neck pain would be less accurate and take longer to respond to images in the left/right neck judgement task than people with no neck pain or people with acute neck pain (Hypothesis 1). For Aim 2, we hypothesised that the usual location of neck pain would affect performance on images showing neck rotation towards their painful side (i.e., people with left-sided neck pain would be less accurate and take longer to respond to images of necks turning to the left and vice versa; Hypothesis 2).

## METHODS

### Participants

The current study is based on data collected as part of a large, online cross-sectional study, from which characteristics of the task and normative data have been published (*Wallwork et al., 2013*). That study collected data from a convenience sample of 1,737 participants, from 40 countries, who were recruited via email, using the Neuro Orthopaedic Institute (NOI, Adelaide, Australia) mailing list, and via social media. Access to a computer with internet capabilities was required to participate. Ethical approval was granted from the University of South Australia Human Research Ethics Committee (Protocol ID HS13-2009). All participants provided informed consent online, as per the Declaration of Helsinki.

### Questionnaires

Participants completed an online questionnaire, providing details on their age, gender, and handedness (*Wallwork et al., 2013*). This study concerns participant responses to questions about neck pain. If a participant reported neck pain, they also answered questions about the duration of their neck pain, the side of neck pain (left, right or bilateral), and whether their neck pain was evoked by neck movement (left, right or bilateral).

### Left/right judgement tasks

Once participants had completed the baseline questionnaire, instructions on how to perform the left/right judgement tasks using the Recognise platform (Neuro Orthopaedic Institute, Adelaide, Australia; www.noigroup.com) were provided. Two sets of images (head/neck; hands) were used for testing and the images in each set were presented in a randomised order. The first set displayed images of people's head and upper torso with their head rotated either towards their left side or their right side (i.e., left and right neck rotation, respectively). The second set displayed images of left and right hands. Participants were instructed to make a judgement on whether the person in the image was rotating their neck to their left or right side, or whether the hand was a left hand or a right hand, respectively. Participants were advised to make a left-sided response by depressing the 'a' key with their left index finger and to make a right-sided response by depressing the 'd' key with their right index finger. Participants were instructed to make

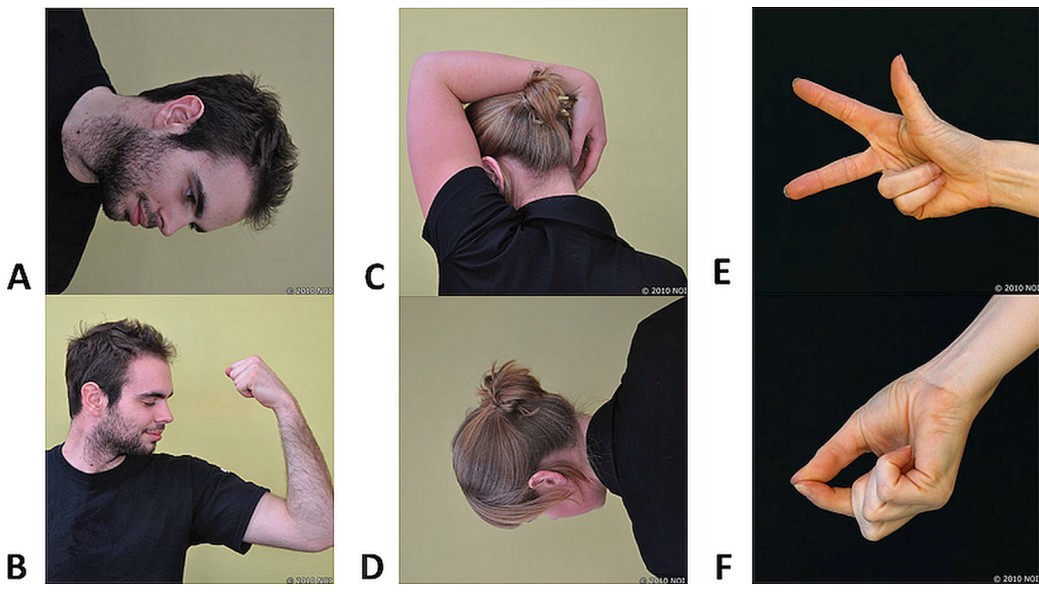

**Figure 1** **Sample images used in the left/right judgement tasks.** (A–D) Head-turning images at various orientations and in various postures. (E and F) Hand images depicting various postures. Photo credit: Juliet Gore; received from Noigroup archived images (www.noigroup.com).

a decision as quickly and as accurately as possible and to avoid guessing. They were informed that they would have a maximum of five seconds per image to respond before the test was automatically advanced and the next image was shown.

Two trial images (either of a left and a right neck rotation or of a left and a right hand) preceded formal testing in order to orientate the participant to the keyboard commands—data were not recorded on these images. The formal test included five left/right judgement tasks, each involving 40 images. The first four left/right judgement tasks displayed images of a head and torso (left/right neck rotation judgements), the fifth task displayed images of hands (left/right hand judgements). The images were randomly presented to be left and right rotation (neck images) or sides (hand images) with images rotated at: 0°, 90°, 180° and 270°. Each task comprised 50% male models and 50% female models, both aged in their early 20's wearing plain black clothing. All left images were mirrored reflection of right images and each participant received all the images (order randomised). See Fig. 1 for sample images.

## Data processing

Left/right judgement performance was analysed using the data from the second and fifth left/right judgement tasks. The first and third tasks were included as methodological controls for task features and the fourth task was included to address a separate question that was unrelated to the aims of the present study. Specifically, the first left/right judgement task (neck images) was included to familiarise participants with the task and to allow for a known learning effect (Boonstra et al., 2012). The third task (identical to task 2) was included as a backup in the situation that people with pain had

difficulty completing task 2 and there were significant missing data. Because there were sufficient data available for task 2, we decided (prior to analysis) to not include data from the third task to minimise potential participant fatigue that was likely to occur given it was the third repetition of neck images. The fourth task was a separate task that included contextual images (with various backgrounds and distractions). Data from the fifth left/right judgement task (hand images) were analysed as a control because the task involved implicit motor imagery of a remote body part, and the use of new images (hands vs. necks) reduced potential for fatigue.

Participants' data were excluded if the second and fifth left/right judgement tasks were not completed in their entirety. Consistent with previous literature, data from images were excluded if the response time was less than 500 ms as this was considered too short of a time to make a judgement response and therefore would likely represent a guess (*Wallwork et al., 2013*). Further, if the response time for eight consecutive images reached this 5 s limit (i.e., the participant timed-out) then the data from those images were excluded because it was assumed the participant was distracted or the internet/computer failed. We included all other responses that were 5 s (i.e., timed-out response) as it was assumed the participant simply took that long to respond or was unsure about that particular response. Last, data sets were excluded where participants did not provide necessary information for covariate analysis (age, gender, handedness) or group allocation (i.e., neck pain). At the completion of data processing, there were 1,404 complete participant datasets.

## Group allocation

Participants were grouped based on: (1) duration of pain (i.e., no pain; pain for less than 3 months considered 'acute pain'; pain for 3 months or more considered 'chronic pain'); (2) location of pain (i.e., left-sided neck pain, right-sided neck pain, bilateral neck pain and no neck pain).

## Statistical analysis

All statistics were performed using SPSS 23.0.0 (SPSS, Chicago, IL, USA). Accuracy and response time data were both tested for normality. Accuracy of responses were not normally distributed and as a result were log transformed. The log transformed accuracy values met normality criteria (as assessed by visual inspection of P–P plots and non-significant Shapiro-Wilk statistic) and were used for all analyses. Thus, for accuracy data, the analysis results were back transformed to provide group specific data (mean, 95% CI). The back transformed mean differences and their 95% CIs were not reported, because the difference between logarithms of two geometric means results in a logarithm of their ratio, not of their difference (*Bland & Altman, 1996*). Age, gender and handedness are known to affect left/right neck judgement performance (*Wallwork et al., 2013*); response time increases with age, is greater in females, and is greater in left-handed people, and accuracy reduces with age. Therefore, these variables were considered in all analyses and included as co-variates where appropriate. Further, a linear regression was

performed using accuracy and response time to assess for a possible speed-accuracy trade-off (i.e., faster performance but incorrect response) which would suggest improper performance of the task. In all analyses, the alpha level was set at 0.05, with a Holm-Bonferroni correction used for all multiple comparisons (*Holm, 1979*).

To determine whether *neck pain and its duration* is associated with impaired left/right judgement performance (Aim 1), univariate ANOVAs were conducted (one each for accuracy and response time) for neck images, with a between-subjects main effect of Pain Duration (no pain, acute pain and chronic pain) and Age as a covariate. If a significant main effect, independent *t*-tests were used to make specific between group comparisons. Identical analyses were completed for left/right judgement performance on hand images (control).

To determine whether the location of neck pain is associated with impaired performance for responses to left-turning and right-turning neck images (Aim 2), accuracy and response time were separately investigated using a 2 (within-subjects main effect of Side of Head Turn in Image: left-sided turning images and right-sided turning images) by 4 (between-subjects main effect of Location of Pain: no pain, left-sided pain, right-sided pain, bilateral pain) repeated measures ANOVA. Such an analysis allowed us to determine if those with bilateral pain were equally impaired for left vs. right images as well as explore whether or not there were task-based features that influenced performance, regardless of the presence or location of neck pain. Therefore, in addition, to specifically evaluate the effect of lateralised pain on performance, repeated measures ANOVAs were completed comparing only those with lateralised neck pain (left-sided neck pain vs. right-sided neck pain) for performance (accuracy and response time) on left-sided turning and right-sided turning neck images.

Given that people with neck pain could have neck pain in one location (i.e., left side neck pain), but experience pain with a movement in the opposite direction (i.e., experience left-sided neck pain when rotating their neck to the right), we ran a sensitivity analysis classifying participants into groups based on the direction of neck rotation which induced their pain. This was completed to ensure that we did not miss a potential effect of the location of neck pain (see Supplemental Information).

## RESULTS

A total of the 1,737 people who completed the online task, 333 were excluded due to incomplete data for the neck pain questionnaires. This resulted in a sample of 1,404 participants from 35 countries. In Test 2, 546 of the 56,160 single responses (i.e., <1% of responses) were eliminated and in Test 5, 601 of the 56,160 single responses (i.e., ~1% of responses) were eliminated due to too short of a response time or eight consecutive timing-out responses. Participant demographics are shown in Table 1.

### Accuracy-response time trade off

People who responded faster were also more accurate ($p < 0.001$, $R^2 = 0.123$). That is, there was no accuracy-response time trade-off.

**Table 1 Demographic information of included participants.** All numbers represent count data unless otherwise specified.

| Demographic variables | All participants | Duration of pain | | | Location of pain | | | | Movement-evoked neck pain | | | |
|---|---|---|---|---|---|---|---|---|---|---|---|---|
| | | No pain | Acute neck pain | Chronic neck pain | No pain | Left | Right | Bilateral | No pain | Left | Right | Bilateral |
| Total | 1,404 | 1,138 | 105 | 161 | 1,104 | 79 | 73 | 96 | 1,075 | 43 | 42 | 118 |
| Male | 422 | 364 | 20 | 38 | 352 | 12 | 17 | 22 | 338 | 9 | 7 | 24 |
| Female | 912 | 720 | 82 | 110 | 701 | 63 | 50 | 68 | 692 | 33 | 32 | 86 |
| Gender not reported | 70 | 54 | 3 | 13 | 51 | 4 | 6 | 6 | 45 | 1 | 3 | 8 |
| Left-handed | 141 | 119 | 9 | 13 | 120 | 9 | 5 | 6 | 112 | 3 | 4 | 11 |
| Right-handed | 1192 | 962 | 92 | 138 | 927 | 66 | 63 | 87 | 913 | 38 | 34 | 101 |
| Ambidextrous | 41 | 33 | 3 | 5 | 33 | 3 | 3 | 1 | 31 | 2 | 2 | 2 |
| Handedness not reported | 30 | 24 | 1 | 5 | 24 | 1 | 2 | 2 | 19 | 0 | 2 | 4 |
| Age (mean ± SD) | 39 ± 12.9 | 37 ± 12.8 | 37 ± 12.5 | 45 ± 12.7 | 37 ± 12.7 | 42 ± 11.8 | 41 ± 13.2 | 42 ±14.0 | 37 ± 12.7 | 42 ± 11.6 | 43 ± 11.6 | 43 ± 13.6 |

## Aim 1: the effect of neck pain and its duration on left/right judgement performance

### Accuracy

*Neck images*

Controlling for age, there was a main effect of Pain Duration ($F_{2,1,400} = 6.36$, $p = 0.002$, partial $\eta^2 = 0.009$) on accuracy of identifying direction of head turn (see Fig. 2A). People with no pain (89.1%, 95% CI [88.1–90.1]) were more accurate ($p = 0.001$) than people with chronic neck pain (83.9%, 95% CI [81.2–86.7]), but were no different ($p = 0.14$) from people with acute neck pain (86.2%, 95% CI [82.9–89.7]). Further, there was no difference in accuracy for judgements of neck images between those with acute neck pain and those with chronic neck pain ($p = 0.28$).

*Hand images*

Controlling for age, there was no main effect of Pain Duration ($F_{2, 1,400} = 0.539$, $p = 0.58$, partial $\eta^2 = 0.001$) on accuracy for left/right judgements of hand images (see Fig. 2B). People with no pain had a mean accuracy score of 87.3% (95% CI [86.7–88.1]), while people with acute and chronic neck pain had a mean accuracy score of 86.3% (95% CI [84.1–88.7]) and 86.7% (95% CI [84.7–88.5]), respectively.

### Response time

*Neck images*

Controlling for age, there was no main effect of Pain Duration ($F_{2, 1,400} = 1.511$, $p = 0.221$, partial $\eta^2 = 0.002$) on response time for judgements of images of necks (see Fig. 2C). People with no pain were no faster at identifying an image of a neck than those with acute pain (mean difference: −61.5 ms, 95% CI [−156.9 to 33.9] ms) or those with chronic pain (mean difference: −53.7 ms, 95% CI [−133.9 to 26.4] ms) and those with acute pain were
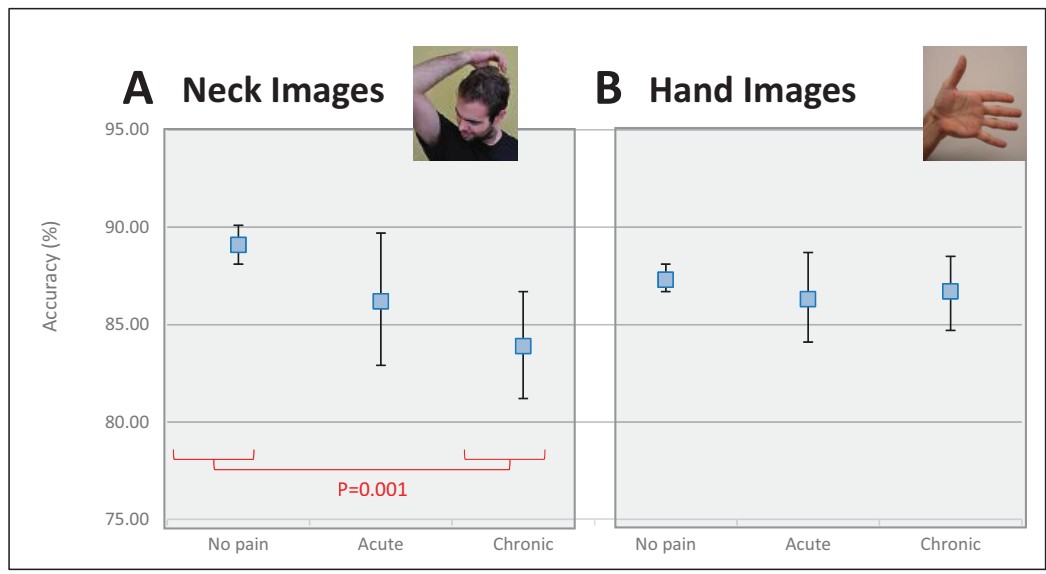

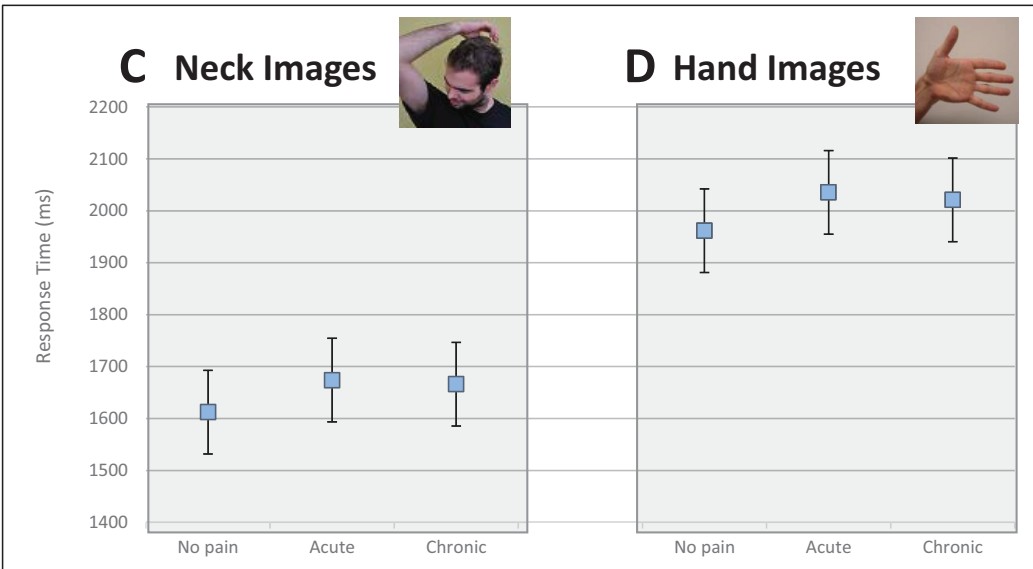

**Figure 2 Effect of pain duration on left/right judgement performance.** (A) Accuracy results for neck images. (B) Accuracy results for hand images. (C) Response time for neck images. (D) Response time for hand images. The red line indicates the significant post-hoc independent *t*-test findings for this comparison performed following an overall main effect of pain duration. Photo credit: Juliet Gore.

no different than those with chronic pain (mean difference: 7.7 ms, 95% CI [−110.4 to 125.9] ms).

*Hand images*

Controlling for age, there was no main effect of Pain Duration ($F_{2,1,400} = 1.491$, $p = 0.225$, partial $\eta^2 = 0.002$) on response time for hand images (see Fig. 2D). People with no pain were no quicker at making judgements than those with acute pain (mean difference: −73.5 ms, 95% CI [−183.6 to 36.5] ms) or those with chronic pain (mean difference: −59.1 ms,

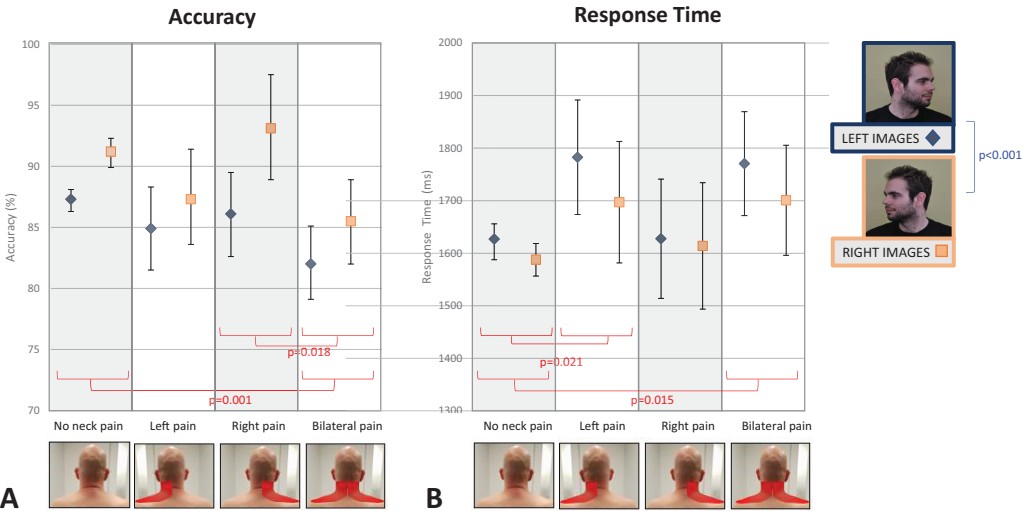

**Figure 3** **The effect of neck pain location (none, left-sided, right-sided, bilateral) on left/right neck judgement performance.** (A) Accuracy. (B) Response time. The blue line shows a main within subject effect of image type (direction of image head rotation). The red lines show significant posthoc independent *t*-test comparisons, following a main significant effect of pain location. Photo credit: Juliet Gore.

95% CI [−151.5 to 33.4] ms), and those with acute pain were no quicker than those with chronic pain (mean difference: 14.5 ms, 95% CI [−121.9 to 150.9] ms).

## Aim 2: the effect of location of neck pain on judgements to left-turning and right-turning neck images

### Accuracy

When considering neck pain in all locations (none, left, right, bilateral), there was a main within-subjects effect of Direction of Image Head Rotation ($F_{1,\ 1,345} = 57.44$, $p < 0.001$, partial $\eta^2 = 0.041$) and a main between-subjects effect of Location of Pain ($F_{3,\ 1,345} = 4.34$; $p = 0.005$, partial $\eta^2 = 0.010$), but no Direction of Image Head Rotation × Location of Pain interaction ($F_{3,\ 1,345} = 2.07$, $p = 0.103$. partial $\eta^2 = 0.005$). Post-hoc analyses revealed that people were less accurate at identifying a left-turning neck than a right-turning neck ($p < 0.001$), that people with no pain were more accurate than people with bilateral pain ($p = 0.001$), and that people with right-sided pain were more accurate than people with bilateral pain ($p = 0.018$; see Fig. 3A).

When comparing accuracy in only those people with lateralised neck pain (i.e., only left- or right-sided neck pain), there was a main effect of Direction of Image Head Rotation ($F_{1,\ 150} = 37.7$, $p < 0.001$, partial $\eta^2 = 0.201$), but no between-subjects effect of Side of Pain ($F_{1,\ 150} = 2.53$; $p = 0.11$, partial $\eta^2 = 0.017$). There was also a significant Direction of Image Head Rotation × Side of Pain interaction ($F_{1,\ 150} = 7.81$, $p = 0.006$, partial $\eta^2 = 0.050$). To explore the interaction effect, post-hoc tests evaluated the effect of image in each group and the effect of group for each image. In regards to the former, post-hoc tests confirmed that both those with left-sided pain and right-sided pain were less accurate for left-turning images than right-turning images ($p = 0.018$ and $p < 0.001$, respectively).

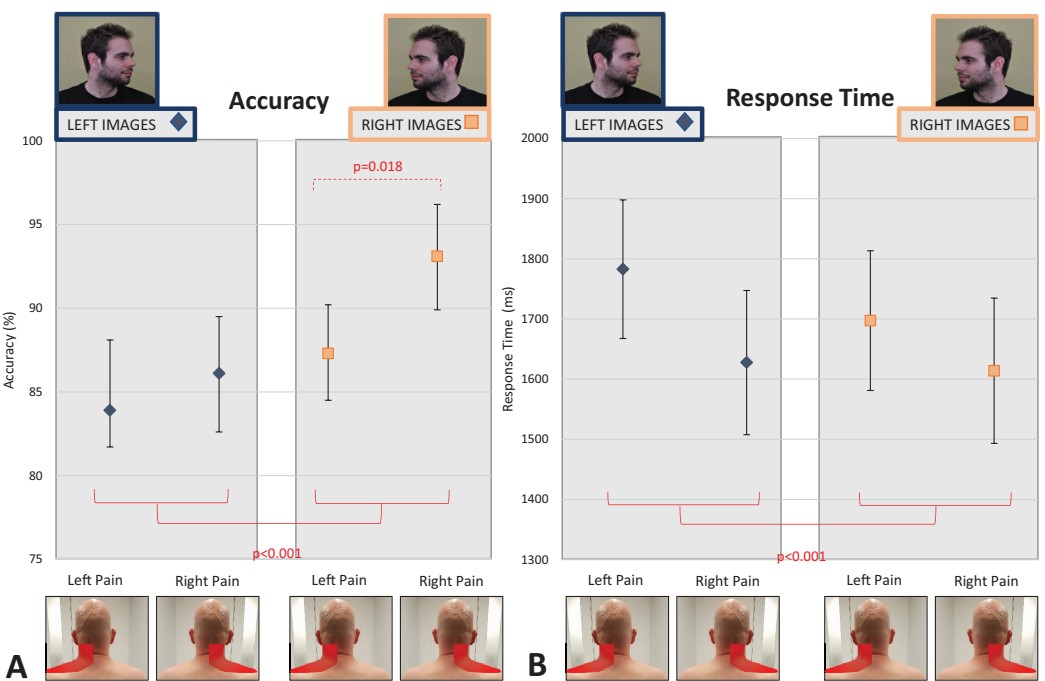

**Figure 4 Effect of unilateral pain location (left- vs. right-sided neck pain) on left/right neck judgement performance.** (A) Accuracy. (B) Response time. The red lines indicate main between group effects of image type (left vs. right head turning images). The dotted red line shows the posthoc independent *t*-test results showing significant performance differences between left- and right-sided pain only for right-sided neck images, following a significant interaction between image type and side of pain. Photo credit: Juliet Gore.

In contrast, post-hoc tests exploring the effect of group for each image show that accuracy was no different between groups for left-turning neck images ($t_{1, 150} = -0.504$, $p = 0.62$), whereas for right-turning neck images those with right-sided neck pain were significantly more accurate ($t_{1, 150} = -2.70$, $p = 0.008$; 93.1%, 95% CI [89.9–96.2]) than those with left-sided neck pain (87.3%, 95% CI [84.5–90.2]; see Fig. 4A). Overall, this suggests that people with right-sided neck pain have a larger difference in accuracy between right- and left-turning images than do people with left-sided neck pain, and this larger difference is driven by superior performance for right-turning neck images.

### Response time

Similar to accuracy findings, there was a main within-subjects effect of Direction of Image Head Rotation ($F_{1, 1,347} = 18.60$, $p < 0.001$, partial $\eta^2 = 0.014$) and a main between-subjects effect of Location of Pain ($F_{3, 1,347} = 3.51$, $p = 0.015$, partial $\eta^2 = 0.008$), but no Direction of Image Head Rotation × Location of Pain interaction ($F_{3, 1,347} = 1.56$, $p = 0.20$, partial $\eta^2 = 0.003$). Post-hoc analyses revealed that people were faster at identifying a right-turning neck image than a left-turning neck image (mean difference = −52.02 ms, 95% CI [−75.68 to −28.36] ms, $p < 0.001$), and that people with no pain were faster at responding to images than people with left-sided neck pain (mean difference = −32.6 ms, 95% CI [−245.4 to −19.8] ms, $p = 0.021$) and people with bilateral neck pain (mean difference = −231.4 ms, 95% CI [−231.3 to 25.2] ms, $p = 0.015$; see Fig. 3B). When only

those with lateralised neck pain (left- or right-sided pain) were included in the analysis, findings were unchanged, although the interaction between Direction of Image Head Rotation × Side of Pain approached significance ($F_{1, 150} = 3.66$, $p = 0.058$, partial $\eta^2 = 0.024$; see Fig. 4B and Supplemental File for full analysis).

### Sensitivity analyses

The results of analyses evaluating the left/right judgement performance effects of neck pain evoked by side-specific movement (e.g., pain evoked with head/neck movement to the left, head/neck movement to the right, both, or neither) were consistent with the typical location of neck pain results reported above. See Supplemental File for full analysis results. Figures S1A (accuracy) and S1B (response time) show results considering full sample (no pain with neck rotation, neck pain with left head rotation, neck pain with right head rotation, neck pain with head rotation in both directions) and Figs. S2A (accuracy) and S2B (response time) show results limited to those with neck pain induced by right or left head rotation.

### Interim summary

Taken together, people were faster and more accurate in identifying a right-turning neck image than a left-turning neck image, regardless of whether or not they experienced neck pain. People with bilateral pain were most impaired—they were less accurate and slower at making neck left/right judgements than those with no pain (and less accurate than those with right-sided pain). People with left-sided pain were slower at identifying neck images than those with no pain, but no less accurate. Last, people with unilateral pain had varying accuracy based on the location of neck pain side: those with right-sided neck pain were more accurate at identifying right-turning neck images than those with left-sided neck pain (but did not differ for left-turning neck images nor for any comparison of reaction time).

## DISCUSSION

We hypothesised that people with chronic neck pain would have impaired performance on a left/right neck rotation judgement task (versus those without neck pain and those with acute neck pain), but not on a left/right hand judgement task (Aim 1). Our hypothesis was partially supported. As hypothesised, people with chronic neck pain were less accurate than people without neck pain on a left/right neck rotation judgement task, but not on a left/right hand judgement task. However, people with chronic neck pain were no less accurate than those with acute neck pain and response time was unaffected by the presence or duration of neck pain. Our second hypothesis, that the location of neck pain would impair performance on images showing neck rotation towards the painful side (Aim 2), was not supported. We found only group differences based on pain location (those with bilateral pain were less accurate and slower than 'no pain' controls, and those with left-sided pain were slower than 'no pain' controls). A significant interaction between side of pain and image (left/right neck rotation) found the opposite of what we predicted: people with right-sided neck pain were more accurate at identifying an image of a right-turning head than those with left-sided neck pain.

Taken together, the current findings support the possibility of impaired neural proprioceptive representation of the neck in people with chronic neck pain, but also show that overall, the impairment is not specific to the painful side of the neck. The distribution or spread of pain may be important, given that those with bilateral pain had consistently worse performance. These results have important theoretical and clinical implications.

**Effect of pain duration on task performance**

There is accumulating evidence that people with chronic pain are less accurate at left/right judgement tasks that use images that correspond to the painful body-part (*Breckenridge et al., 2018*), suggesting disruptions to neural proprioceptive representations that are associated with movement. Our findings are consistent with this body of literature and specifically support previous work in people with recurrent, idiopathic neck pain that showed lower accuracy at a left/right neck judgement task than that observed in people without neck pain (*Elsig et al., 2014*). That similar impairments in accuracy were not detected in one small study in people with chronic WAD (*Pedler, Motlagh & Sterling, 2013*), might suggest that differences exist within people with neck pain, i.e., between those with and without WAD. However, there are two issues that suggest otherwise. Firstly, the study by *Pedler, Motlagh & Sterling (2013)* involved a different task—judging whether an image showed the left or right side of the neck, in contrast to our task—judging whether an image showed the neck and head rotated to the left or the right. It seems probable that the tasks interrogate different processes. Further, it was not powered to detect a difference (only 24 participants in the control group), but instead powered to detect a correlation between performance in left/right judgements and factors that are thought to be predictive of non-recovery (*Pedler, Motlagh & Sterling, 2013*). Secondly, it is highly likely that our sample included a large number of people with WAD, but our online design meant we could not assess them, nor definitively classify people as having WAD or not.

Reduced accuracy in the neck left/right judgement task, but not in the hand task, supports the somatotopically specific nature of impairments in proprioceptive representation. This specificity of impairment is largely consistent with past research in several chronic pain conditions, including back pain (*Bowering et al., 2014*) and arm pain (*Schwoebel et al., 2001*; *Moseley, 2004b*) (see also systematic review findings: (*Breckenridge et al., 2018*)), and also in experimentally induced pain (*Moseley et al., 2005*) and even the expectation of experimentally induced pain (*Hudson et al., 2006*). It is important to consider the wider body of evidence when interpreting the current results because different problems might underpin impairment of accuracy as distinct from response time. Full review is beyond the scope of this paper, but deficits in accuracy are more consistent with disrupted proprioceptive representation and deficits in RT are more consistent with unequal weighting between representations of either hand, or movement (see (*Moseley et al., 2012*; *Wallwork et al., 2016*; *Wallwork, Bellan & Moseley, 2017*) for extensive reviews, although see *Pelletier et al. (2018)* for other possible contributors to performance).
Recent findings by Pelletier and colleagues have shown that left/right judgement performance is related to both sensorimotor and cognitive function in people with pain (*Pelletier et al., 2018*). That is, performance on tasks including tactile acuity and motor function were related to left/right judgement accuracy of the affected body part, as was taking pain medication. Additionally, poor performance in a cognitive stroop task was associated with general impaired left/right judgement performance. Thus, it is possible that the differences observed between the chronic pain and no pain groups in our study (Aim 1) could, in part, be attributable to other factors (such as those reported by Pelletier and colleagues) than to impairments to the proprioceptive neural representation. However, these relationships between sensorimotor function and left/right judgement performance are not straightforward: in people with chronic painful knee osteoarthritis, impairments in tactile acuity were not related to left/right judgement performance (*Stanton et al., 2013*). Also, if impaired cognitive function in the chronic neck pain group influenced their task performance (as per *Pelletier et al., 2018*), we would expect impaired performance on both hand and neck image left/right judgement tasks, which we did not see. Further work is clearly required to understand the nuanced and complex mechanisms contributing to performance in this task.

That people with neck pain have impaired accuracy on left/right judgements for neck images has particular clinical relevance, because it raises the possibility that treatments aiming to reverse that deficit might help to reduce pain in those with chronic neck pain. Certainly, in other pain populations (i.e., complex regional pain syndrome and phantom limb pain), motor imagery training, which includes left/right body part judgements, has been shown to be effective at reducing pain (*Moseley, 2004a*, *2006*). There are three important caveats here that urge caution before adding motor imagery training to a clinical toolbox. Firstly, it is not known whether an accuracy deficit of ~6% is clinically meaningful. Secondly, whether deficits vary according to diagnosis or mechanism of injury remains to be determined. Thirdly, any recommendations for new treatments should only be made once the treatment is known to be safe and effective, as determined in clinical trials.

That we found no difference in accuracy between the acute pain group and the healthy control group (without neck pain) supports the idea that these impairments do not occur in the acute stages. Such findings are largely consistent with previous work in a back pain sample showing that impairments are minimal in people with current back pain (but no history of back pain) (*Bowering et al., 2014*). However, we also found that the acute pain group did not differ in accuracy to the chronic pain group either which raises the possibility that small changes in performance may occur in the acute phase, but they are not large enough to be detected even in a large cohort such as this, which implies any difference would be very small and likely to be unimportant. Investigating longitudinal change in proprioceptive representation as pain persists appears warranted because perhaps early deficit is a risk factor for poor recovery.

While left/right judgement accuracy was impaired, response time was not. This may be explained by the different processes underpinning the two outcomes. That is, RT deficits in limb pain are currently interpreted as reflecting unequal weighting of space-based or

side-based representations (see above). There is increasing evidence showing the presence of space-based deficits in processing in association with limb pain, with a range of effects including thermoregulation, motor control and tactile processing (*Moseley, Gallace & Spence, 2009*; *Stanton et al., 2012*; *Reid et al., 2016*, *2018*) (see *Moseley, Gallace & Spence (2012)* for a review). However, in limb pain, the spatial representations seem to involve the area of peripersonal space in which the limb is used. This clearly is not applicable to spinal pain, so it is perhaps unsurprising that we did not observe RT deficits here.

## Effect of location/side specific pain on performance

Our finding that location specific neck pain (Aim 2) or movement-evoked neck pain (see Supplemental Material) did not affect performance to images showing neck rotation to their affected side would not be predicted on the presumed relationship between proprioceptive representation and motor imagery performance (*Bray & Moseley, 2011*; *Schmid & Coppieters, 2012*; *Wallwork et al., 2016*). That is, we predicted that performance on a task that requires access to the representations for a body part (or movement) that is normally associated as being painful would be worse than for a task that requires access to the representations for a body part (or movement) that is not normally associated as being painful. Such a hypothesis is supported by past work in people with painful osteoarthritis (*Stanton et al., 2012*) and also by contemporary theories (i.e., the cortical body matrix theory (*Moseley, Gallace & Spence, 2012*)) in this area.

Contemporary theories of motor processing emphasise the distributed nature of processing and the conceptual construct of neural networks (or 'neurotags' (*Butler & Moseley, 2003*; *Moseley & Butler, 2015*)) that are under the influence of a potentially infinite number of other neural networks (see *Wallwork et al. (2016)* for a review). For example, a movement of neck rotation (the 'output'), depends on incoming data (including proprioceptive cues, visual cues (head rotation), tactile cues (from skin stretching)), stored data (past experience) and predicted outcomes of a movement. As part of this complex process, motor efferent copies (i.e., a copy of the motor command) are created with multi-sensory feedback from the movement used to determine if the movement occurred as planned. The more often the network is activated, the stronger and more precise it becomes (*Pearson, Finkel & Edelman, 1987*; *Pilz et al., 2004*), which in turn facilitates activation of the 'learned' network. It might be predicted then, that when performing a left/right neck rotation judgement task that uses these disrupted proprioceptive representations, performance would be specifically hindered for an image congruent with painful movement—even when the full movement output has not occurred (i.e., they have not actually moved). That we did not consistently find an interaction effect in this study indicates that perhaps these proprioceptive representations have general alterations for the affected body part, and are not specific to the location of pain, or movement-evoked side of pain. Alternatively, it may be that the precision with which neck proprioceptive input is represented for side-specific movements is less than we would expect in lateralised body parts (e.g., left and right hand).

We found that people were generally more accurate at identifying images of right neck rotation than left neck rotation, regardless of neck pain status. Previous studies have

identified side-specific effects as a function of hand dominance: right-handers are more accurate at identifying right hand images than left-handers (Braithwaite et al., in preparation; *Wallwork et al., 2013*) and perhaps these findings transfer to our results too. In our sample, 1,192 of 1,404 were right-handed so there was a much stronger right-handed presence in our sample group which would be consistent with this idea. However, it seems unlikely that the same effect of dominance extends to the neck like it does the hand. Of relevance, people are faster and more accurate at a left/right judgement task when images are rotated 90° clockwise (medial) than when they are rotated 90° anti-clockwise (lateral) for images of the neck (*Wallwork et al., 2013*) and for images of the back ($n = 1,189$ participants) (*Bowering et al., 2014*). Given that past work has also shown that such differences based on image orientation may reflect a switch from implicit motor imagery to more proprioceptive-visual matching strategies (as seen in hand left/right judgements) (*De Simone et al., 2013*), these hypotheses clearly require further investigation.

An interaction between side of the body and side of pain was seen only for analyses considering the location of neck pain (left vs. right-sided), but it was opposite to the direction hypothesised. People with right-sided neck pain were more accurate at identifying an image of a right-turning head than people with left-sided neck pain. While previous work by *Stanton et al. (2012)* found that people with right-sided knee osteoarthritic pain or upper limb pain were generally more accurate than those with left-sided pain at identifying images of hands or feet, enhanced performance for right-sided pain × right sided images was not seen. Specifically, those with right-sided knee or upper limb pain were no more accurate at identifying a right hand/foot than a left hand/foot, whereas in participants with left-sided pain, performance was most impaired for left hand or foot images vs. right images (side of pain matched side of impairment) (*Stanton et al., 2012*). Together with generalised reduction in performance for left vs. right-sided neck images seen here, such findings raise the possibility that there might be a lateralised performance effect whereby spatial lateralisation of processing during the task (i.e., in the right posterior parietal cortex) results in an interference effect with left-sided (right processed) proprioceptive representations (i.e., left images), which can be further affected by left-sided (right processed) pain, but is relatively spared by right-sided (left processed) pain. However, why performance for right images in people with right-sided neck pain is seemingly enhanced remains unclear. To our knowledge, this line of enquiry remains to be tested.

Lastly, we found that people with bilateral neck pain (typical and movement-evoked) were slower and less accurate than people with no neck pain. Such findings raise the possibility that the disruptions to proprioceptive representation are greater when pain is more widespread in area (i.e., bilateral). Finally, people with right-sided neck pain were more accurate than people with bilateral neck pain, but no faster than people with left-sided pain, and people with no pain were also faster than people with left-sided pain. This pattern of findings requires validation, but generally supports that the largest impairments are seen in those with bilateral pain, followed by left-sided pain, and the least impairment in those with right-sided pain.
### Study limitations

While this is one of the largest left/right judgement studies undertaken to date, interpretation of our results should consider several limitations. Firstly, the online nature of the study means that computer malfunctions, screen size, screen resolution, and refresh rate could increase variance and reduce our power, although it is unlikely to systematically bias the results. Furthermore, being an online study, we cannot exclude that participants took part more than once on different devices or provided inaccurate information when answering the online questionnaire. To combat limitations imposed by this increase in variability of the data, we purposefully aimed to collect a large sample. There is also the possibility that participants' responses to the fifth block (hand images) may have been affected by fatigue when compared to responses to the second block (neck images). Although possible, we think this is unlikely as fatigue is primarily seen when an identical task is repeated. Because the hand task in the fifth block was new, we anticipate that fatigue is less of a concern. If fatigue were a factor, we would expect that people with neck pain would perform worse at the hand task than pain-free people (*Dailey, Keffala & Sluka, 2015*), which was not observed. Additionally, we were unable to explore image orientation specific performance results for neck and hand images; these data were not retrievable. Given that evaluating image orientation can help to interpret what might underlie accuracy impairments (e.g., impairments in motor imagery or alterations in visual processing (*Edwards et al., 2019*)), future research that purposefully explores the influence of neck image orientation on performance is clearly needed.

Given an online study, there are also inherent limitations in participant self-report for 'acute' and 'chronic' neck pain, because different interpretations of duration may exist for different people. For example, despite providing clear descriptors about duration, it is possible that someone with an ongoing chronic problem might report 'acute' pain because they are suffering a recent flare-up. Such challenges may also contribute to the lack of performance difference seen between acute and chronic neck pain. Furthermore, while it would have been of interest to consider pain intensity as a covariate in the analyses, limitations in the online questionnaire meant that pain intensity values were only collected for those who reported current neck pain (e.g., but not those who might have acute or chronic neck pain, but no neck pain at the exact time of testing). Unfortunately then, including pain as a covariate would have excluded numerous participants from the analysis. Additionally, given that past studies have shown an inconsistent relationship between pain intensity and left/right judgement performance (*Bray & Moseley, 2011*; *Linder, Michaelson & Roijezon, 2016*; *Pelletier et al., 2018*; *Stanton et al., 2018*), this makes the relevance of including pain intensity as a covariate less certain. For these reasons pain intensity was not included in our analyses. Lastly, as reported in a previous study using these data (*Wallwork et al., 2013*), using the 'a' and 'd' keys on the keyboard as a response apparatus has limitations in that both keys are located on the left of the keyboard (i.e., in the left side of space). We now understand that a bias in the allocation of attention to a spatially-defined location could influence responding (*Moseley, Gallagher & Gallace, 2012*; *Reid et al., 2016*); however such influences are less likely to impact our main findings, given

that the neck is not like the limbs, which are normally situated in one spatial field (right or left of body midline). Regardless, future work in this area should aim to use keys that result in the response hands being positioned on either side of the body midline. Finally, that we did not lodge and lock our experimental protocol prior to conducting this experiment reflects that we began the study before we fully understood the importance of doing so. This practice enhances the transparency of research and is now recommended for observational and clinical designs in many fields including pain (*Lee et al., 2018*; *Cashin et al., 2020*).

## CONCLUSIONS

In our large online cohort ($n$ = 1,404), including 266 people with neck pain, we found evidence of disruptions to proprioceptive representations that were specific to the neck in people with chronic neck pain. In particular, people with chronic neck pain were less accurate at a left/right neck rotation judgement task, but not a left/right hand judgement task, than people with no neck pain. Contrary to our expectations, we found that the location of neck pain generally did not influence responses to images associated with movement towards their painful side. Overall these findings indicate that investigation of a graded motor imagery programme to reduce pain in people with neck pain may be warranted. However, further investigations in neck pain subgroups, such as WAD, need to be addressed.

### Funding

The authors received no funding for this work.

### Competing Interests

In the last 5 years, G. Lorimer Moseley has received support from: ConnectHealth UK, Seqirus, Kaiser Permanente, Workers' Compensation Boards in Australia, Europe and North America, AIA Australia, the International Olympic Committee, Port Adelaide Football Club and Arsenal Football Club. Professional and scientific bodies have reimbursed him for travel costs related to presentation of research on pain at scientific conferences/symposia. He has received speaker fees for lectures on pain and rehabilitation. He receives book royalties from NOIgroup publications, Dancing Giraffe Press & OPTP. G. Lorimer Moseley is an Academic Editor for PeerJ. Tasha R Stanton has received support from Eli Lilly Ltd for travel and accommodation costs.

### Author Contributions

- Sarah B. Wallwork conceived and designed the experiments, performed the experiments, analysed the data, prepared figures and/or tables, authored or reviewed drafts of the paper, and approved the final draft.
- Hayley B. Leake performed the experiments, analysed the data, prepared figures and/or tables, authored or reviewed drafts of the paper, and approved the final draft.

# PeerJ

- Aimie Peek analysed the data, prepared figures and/or tables, authored or reviewed drafts of the paper, and approved the final draft.
- G. Lorimer Moseley conceived and designed the experiments, performed the experiments, analysed the data, prepared figures and/or tables, authored or reviewed drafts of the paper, and approved the final draft.
- Tasha R. Stanton conceived and designed the experiments, analysed the data, prepared figures and/or tables, authored or reviewed drafts of the paper, and approved the final draft.

## Human Ethics

The following information was supplied relating to ethical approvals (i.e. approving body and any reference numbers):

The University of South Australia granted ethical approval to carry out the study within its facilities (Ethical Application Ref: HS13-2009).

## Data Availability

The raw data is available in the Supplemental Files.

## Supplemental Information

Supplemental information for this article can be found online at http://dx.doi.org/10.7717/peerj.8553#supplemental-information.

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
