# Peer review of "Implicit motor imagery performance is impaired in people with chronic, but not acute, neck pain"

_PeerJ, doi:10.7717/peerj.8553_

## Round 0.1 · original submission · Major Revisions

I have now heard back from two expert reviewers. Both find the paper interesting but additional work still needs to be done in order to satisfactorily address the concerns raised in their reports. I invite you to address all the issues raised by the reviewers in a major revision of the manuscript.

Reviewer 1 ·

Basic reporting

no comment

Experimental design

no comment

Validity of the findings

no comment

Additional comments

The authors reported from an on-line study accuracy and reaction time of 1404 individuals with and without neck pain for two (neck and hand) laterality judgement tasks. They performed different analyses when grouping participants under 3 status (neck pain status, pain location and head rotation-evoked pain). Overall, individuals with chronic neck pain were less accurate but not slower than individuals without pain (main effect of neck pain status). When grouped with pain location and head rotation-evoked, the results are more disparate. The authors concluded that individuals with chronic neck pain have disrupted proprioceptive representations, without an influence of pain location and movement-evoked pain.
Please find below general concerns and minor comments:

Abstract
- The results when individuals were grouped on pain location status are confusing. The authors should only present main findings that help understanding the general message. Also, the authors should clearly identify the different analyses depending on the grouping status to help the reader to better understand the results (for example, after ‘Of 1404 participants, 105 reported acute neck pain and 161 reported chronic neck pain’, they could write ‘When grouped on neck pain status, people with […]. When grouped on pain location, […]’ or something equivalent).
- l.50: When saying ‘People were faster and more accurate’, does this mean that a test was performed for accuracy and another for RTs? However, only one p-value was reported (p<0.001).

Introduction
- The authors could provide further information to understand the relevance of having Aim 2 and Aim 3, especially when results of aim 2 and 3 are (almost) similar?
- l.137-139: the hypotheses are not in line with the Aims as the authors did not talk about comparison with hand left/right judgement task (l.133-137).

Method
- l.198-203: while the comparison between neck and hand laterality judgement tasks is relevant, how did the authors deal with the fatigue, as mentioned l.201 (the hand tasks being the fifth block)?
- l.210-211: ‘if the response time for eight consecutive images […] data from those images were excluded’. Why eight consecutive images, and not each time the trial lasts 5 seconds?

Statistical analyses
- It said line l.235 that an ANOVA was performed to assess for a possible speed-accuracy trade-off. What was the variable tested?

Results
- l.329-340: the main message for these results is not clearly presented. The authors should try to reformulate this paragraph, only presenting the significant results of the interaction Direction of Image Head Rotation x Side of Pain interaction.
- l.361-362 ‘There were no clear location-specific effects on performance – only differences based on pain location were found’. This seems contradictory.
Why was the analysis on hand laterality judgement task only presented for Aim1?

Discussion
- Did the authors ever consider the intensity of neck pain as a possible co-variate to explain the results? This could be mentioned in the limitation section of the discussion.


Minor comments
- l.226-227: were accuracy responses normally distributed after log transformation?

Reviewer 2 ·

Basic reporting

The ms is generally well-written and referenced. A few minor comments:
Line 518: What specific theories?

The authors state “motor efferent copies (i.e., a copy of the motor command sent) are created and used to determine if the movement output occurred as planned. Theories with which I am familiar typically assume that efferent copy underlies the “forward model” and allow prediction whereas sensory feedback permits one to determine if the movement occurred as planned. Please clarify this.

In line 303 the authors refer to performance at “identifying an image of a neck” rather than identifying the direction of head/neck turn; this should be corrected.

Experimental design

The design is generally adequate but several points should be addressed.

The absence of information about pain severity is striking and a significant limitation. It is certainly possible that pain severity rather than chronicity or “location” of pain is the primary determinant of performance. Pain severity ratings are always subjective and potentially confounded by other factors, but it is not clear that the ratings would be more unreliable if provided on-line than in an interview setting.

Why were 4 blocks of data collected but only the second and fifth blocks reported in the manuscript? If fatigue was anticipated, why were the 3 and 4th blocks included? Was there evidence from behavioral measures (e.g., RT) to suggest that fatigue was an issue? If so, what are the implications for the hand laterality data? Were the results different if all of the data were included?

How much data was eliminated because of timing out or too short responses?

Validity of the findings

The findings appear solid and the analyses generally well motivated.

The major weakness of the manuscript in my view is the discussion. The authors conclude that the data “support the presence of impaired neural proprioceptive representation of the neck in people with chronic neck pain”. This conclusion is not adequately supported. Nor is the self-referential conclusion that deficits in accuracy are more consistent with proprioceptive deficit and deficits in RT are more consistent with unequal weighting. Although one wouldn’t expect an exposition similar to that presented in the reviews cited, this claim needs to be developed; as the discussion is long and at some points rambling (e.g., speculation about the absence of effects of side of pain), there is plenty of space to present evidence relevant to this point.

From the early work of Parsons to the present, many investigators have drawn conclusions based on differences in RT as a function of stimulus type (e.g., the degree of angular deviation of the stimulus). As these knowledgeable authors well, know, with normal subjects there is a very characteristic RT pattern for hand and foot identification tasks, presumably reflecting the time taken to mentally rotate the body part to match the pictured stimulus. RTs and accuracy information should be provided for each type of stimulus (e.g., 60° right head turning; left hand palm up at 270°) rather than just summary measures collapsed across direction and magnitude of head turning. Such an analysis may be useful in addressing the question of a relationship between neck position and location of pain, for example.

Additional comments

In general, I thought this to be an interesting and generally well-done manuscript.

---

## Round 0.2 · accepted · Accept

The comments of both reviewers were addressed adequately. There are no further comments.

Reviewer 1 ·

Basic reporting

no comment

Experimental design

no comment

Validity of the findings

no comment

Additional comments

The authors responded point-by-point to my comments.
I have no further remark.